# Is It Possible to Differentiate *Pneumocystis jirovecii* Pneumonia and Colonization in the Immunocompromised Patients with Pneumonia?

**DOI:** 10.3390/jof7121036

**Published:** 2021-12-02

**Authors:** Yudy A. Aguilar, Zulma Vanessa Rueda, María Angélica Maya, Cristian Vera, Jenniffer Rodiño, Carlos Muskus, Lázaro A. Vélez

**Affiliations:** 1Grupo Investigador de Problemas en Enfermedades Infecciosas-GRIPE, Facultad de Medicina, Universidad de Antioquia, Medellín 050031, Colombia; mangelicamaya@gmail.com (M.A.M.); jmedarom_12@yahoo.es (J.R.); lazarovelezg@gmail.com (L.A.V.); 2Facultad de Medicina, Universidad Pontificia Bolivariana, Medellín 050031, Colombia; zulmaruedav@gmail.com (Z.V.R.); cristian.vera.marin@hotmail.com (C.V.); 3Department of Medical Microbiology and Infectious Diseases, University of Manitoba, Winnipeg, MB R3E 0J9, Canada; 4Section of Infectious Diseases, Hospital Universitario San Vicente Fundación, Medellín 050010, Colombia; 5Unidad de Biología Molecular y Computacional, Programa de Estudio y Control de Enfermedades Tropicales-PECET, Facultad de Medicina, Universidad de Antioquia, Medellín 050010, Colombia; carmusk@yahoo.com

**Keywords:** quantitative real time PCR, *Pneumocystis jirovecii*, pneumonia, colonization, bronchoalveolar lavage (BAL), oropharyngeal washes (OW)

## Abstract

Respiratory sample staining is a standard tool used to diagnose *Pneumocystis jirovecii* pneumonia (PjP). Although molecular tests are more sensitive, their interpretation can be difficult due to the potential of colonization. We aimed to validate a *Pneumocystis jirovecii* (Pj) real-time PCR (qPCR) assay in bronchoscopic bronchoalveolar lavage (BAL) and oropharyngeal washes (OW). We included 158 immunosuppressed patients with pneumonia, 35 lung cancer patients who underwent BAL, and 20 healthy individuals. We used a SYBR green qPCR assay to look for a 103 bp fragment of the Pj *mtLSU rRNA* gene in BAL and OW. We calculated the qPCR cut-off as well as the analytical and diagnostic characteristics. The qPCR was positive in 67.8% of BAL samples from the immunocompromised patients. The established cut-off for discriminating between disease and colonization was Ct 24.53 for BAL samples. In the immunosuppressed group, qPCR detected all 25 microscopy-positive PjP cases, plus three additional cases. Pj colonization in the immunocompromised group was 66.2%, while in the cancer group, colonization rates were 48%. qPCR was ineffective at diagnosing PjP in the OW samples. This new qPCR allowed for reliable diagnosis of PjP, and differentiation between PjP disease and colonization in BAL of immunocompromised patients with pneumonia.

## 1. Introduction

*Pneumocystis jirovecii* pneumonia (PjP) affects patients with severe immunosuppression, such as those with an acquired immunodeficiency syndrome (AIDS), autoimmune diseases of the connective tissue, and hematological malignancies, as well as the organ transplant recipients. Although PjP can be lethal in up to 80% of cases [1], most patients survive when treated with trimethoprim–sulfamethoxazole (TMS/SMX) [2]. Diagnosis of PjP is based on a microscopic visualization of the fungus in respiratory samples (from both bronchoalveolar lavage [BAL] and induced/spontaneous sputum) using special stains such as methenamine silver, toluidine blue O (TBO), or direct fluorescent antibody (DFA). A positive microscopy for *P. jirovecii* (Pj) in respiratory secretions is sufficient for the diagnosis of PjP [3].

Introduction of molecular techniques has allowed for the detection of Pj genetic material in invasive and non-invasive respiratory samples, such as oropharyngeal washes (OW) [4,5,6], even in asymptomatic and healthy individuals, demonstrating that a positive result can be indicative of either colonization or disease [7,8,9].

However, when considering the pathophysiology of PjP, it is logical to assume that a high respiratory fungal load suggests a disease since lung damage and symptoms largely depend on a fungal load present within the alveoli. It has been shown that increased *Pneumocystis* load produces pro-inflammatory cytokines and surfactant proteins that change the surface tension and lead to a gas exchange failure and respiratory deterioration of the patient [10,11]. Consequently, techniques that quantify *Pneumocystis* load, such as quantitative PCR (qPCR), would help establish a threshold for discriminating between disease and colonization. However, cut-off values reported thus far in qPCR protocols often generate indeterminate test results due to the overlapping values [4,12,13,14,15,16,17,18,19,20,21,22,23,24,25,26]. These inconsistencies may be partially attributed to the differences among the immunocompromised study populations, in addition to the technical aspects associated with the qPCR protocols (i.e., different target sequences, qPCR platforms, detection or extraction methods, amount of DNA in a sample, normalization of samples with a simultaneous amplification of human genes), or to the way the results are reported.

On the other hand, the term colonization (also known as “carrier”) has been used when a fungus is detected by the nucleic acid amplification techniques in asymptomatic individuals [27]. However, symptomatic respiratory patients with tuberculosis, systemic mycosis, and other pulmonary diseases may also become colonized by Pj or other potential pathogens. Thus, to appropriately interpret a positive result among these individuals, quantitative techniques, such as qPCR, need to consider that a positive qPCR result may not necessarily mean PjP.

Based on the aforementioned limitations, we aimed to clinically validate a qPCR assay that allows for PjP diagnosis, and discrimination between colonization and disease (PjP) in immunocompromised patients with suspected PjP. We also described their characteristics and follow-up according to Pj status.

## 2. Materials and Methods

### 2.1. Study Design

The study was a prospective cohort study.

### 2.2. Inclusion and Exclusion Criteria

The following groups were prospectively recruited:

Group 1: Immunocompromised patients with pneumonia between June 2007–January 2010. We hired a full-time clinician that went every day to the emergency room to identify all people who were admitted to the two hospitals with a suspicion of pneumonia. Then, she reviewed the clinical chart, contacted the patient, and if the patient met the inclusion criteria and none of the exclusion criteria, she enrolled the patient in the study and took the OW. Only a single episode of pneumonia was included per patient. We collected bronchoscopic BAL and OW samples from each patient.

Follow-up: Until discharge or death. During hospitalization, we recorded information on patient demographics, past clinical history, clinical information for the current episode, laboratory parameters, and the presence of mycobacteria, fungi, and pyogenic bacteria, in both BAL and blood samples. After hospitalization, each participant was followed up with at 6 and 12 months by phone call, to ask about new hospitalizations (date(s) and reason for hospitalization(s)), survival, antiretroviral treatment, and prophylaxis with TMS/SMX. We also reviewed the clinical chart to identify if the patient was hospitalized (date(s) and reason for hospitalization(s)) during the year of follow-up and the survival.

Group 2: Patients with lung cancer (a known risk factor for Pj colonization [28]) who had undergone BAL to confirm the diagnosis of neoplasia were included between October 2007 and March 2010, and had BAL and OW samples collected.

Group 3: Immunocompetent individuals (blood donors) aged 18 and 65 years old, who had no known comorbidities or respiratory symptoms, and who, according to the literature, had a low risk of colonization by Pj [29,30,31,32]. This group was recruited between November 2010 and February 2011 and had OW samples collected. Groups 2 and 3 were chosen to evaluate qPCR in people with differing risks of Pj colonization.

Details of the inclusion and exclusion criteria are listed in Table 1.

### 2.3. Setting

The study was set in two tertiary institutions (Hospital San Vicente Fundación and Hospital La María) in Medellín, Colombia. 

### 2.4. Collection of Respiratory Samples and Microscopy

Each bronchoscopic BAL sample was obtained according to the standardized protocols [33] at each institution, and 5 to 20 mL of the sample was sent to our laboratory. OW was taken within the first 48 h of BAL. After oral washing with a sterile saline solution to eliminate any food remnants, OW was performed using 25 mL of 0.9% saline solution for 60 s [34]. To increase cell recovery, an oropharyngeal rayon tip swab (COPAN DIAGNOSTICS^®^, Corona, CA, USA) was collected from each patient and placed in a container with the OW sample. 

Volumes of 3–5 mL of the BAL samples and 10 mL of the OW samples were cytocentrifuged within the first 24 h and used to prepare TBO and DFA slides, as described by Rodiño et al. [35]. The presence of human cells was microscopically verified in all the samples. The remaining sample was coded for blind processing and stored at −80 °C for DNA extraction.

### 2.5. Definitions

*Pneumocystis jirovecii* pneumonia was defined as:

Confirmed: The patient had pneumonia as defined in Table 1 and had a positive result by TBO and/or DFA in the BAL sample.

Probable: A positive qPCR below the cut-off established as a threshold for PjP diagnosis, the patient had an interstitial pneumonia (pneumonia as defined in Table 1 and bilateral interstitial infiltrates in the chest X-ray or ground-glass pattern in the high-resolution computed tomography), with a negative result by TBO and DFA in the BAL sample, with no other microbiological diagnosis that explained the pneumonia, and had clinical improvement with treatment dose of PjP.

The research team did not participate and were not involved in any clinical decision from the infectious disease specialists or pulmonologists in charge of the patient. The research laboratory delivered the TBO and DFA results to the clinicians in charge of the patients immediately after the BAL samples were processed. As the qPCR was under development, we did not provide any qPCR result to the clinicians.

### 2.6. DNA Extraction Protocol

After gradually thawing the samples, 2 mL of the BAL samples and 10 mL of the OW samples were centrifuged at 4500 r.p.m. for 20 min at 4 °C. The pellet was resuspended in 200 µL of the same supernatant. DNA extraction was performed according to modified DNeasy Blood & Tissue Kit protocol instructions (QIAGEN^®^, Hilden, Germany), with the proteinase K and AL buffers added prior to incubation at 56 °C.

### 2.7. qPCR Assay

For the real-time PCR assay that quantified Pj, we amplified a fragment of the *mitochondrial large subunit ribosomal RNA (mtLSU rRNA)* gene [36]. The qPCR targeting the 108 bp mtLSU rRNA region was performed in 25 uL containing 0.3 µM of each primer (PJ mtLSU Forw (5′- AAGGGAAACAGCCCAGAACA-3′) and PJ mtLSU Rev (5′- CTGTTTCCAAGCCCACTTCTT-3′), 1X SYBR-green QuantiFast (QIAGEN^®^, Hilden, Germany)) and 5 µL of DNA from the samples or plasmid. All reactions were conducted in a SmartCycler thermocycler (Cepheid^®^, Sunnyvale, CA, USA) employing an initial cycle of 10 min at 95 °C, followed by 37 cycles each at 95 °C for 30 s, 65 °C for 30 s, and 72 °C for 20 s. A positive and specific reaction was defined as a threshold cycle (Ct) between 10 and 36.99 and a Tm of 77 °C ± 2 °C. A linear dynamic range of qPCR between 50 and 2 × 10^8^ copies/µL had high reproducibility (CV < 4%), amplification efficiencies close to 100% (slopes of −3.2 ($\bar{x}$: 3.24 ± 0.02)), and standard curves with coefficients of determination (R^2^) higher than 0.98 ($\bar{x}$: 0.996 ± 0.01).

Positive and negative controls were included in all qPCR assays. The efficiency of each positive reaction was determined using LinReg software [37]. If samples did not amplify, the presence of inhibitors was ruled out by amplifying a fragment of β-globin or glyceraldehyde-3-phosphate dehydrogenase (GAPDH) human genes by conventional PCR. All procedures and sample handling were conducted according to Clinical and Laboratory Standards Institute (CLSI) [38,39,40] and the European molecular diagnostic guidelines [41].

### 2.8. Sample Size and Study Population

A sample size of 133 OW was estimated based on (a) PjP prevalence of 18% among severely immunocompromised patients with pneumonia [33], (b) 95% power, (c) 99% specificity for microscopy (the test reference), and (d) a specificity of 85% in OW [5] for qPCR when the MSG gene was amplified amplifying the major surface glycoprotein (MSG) gene.

### 2.9. Statistical Analysis

Data were analyzed using the SPSS version 26.0 (SPSS^®^ Inc., Chicago, IL, USA) and Epidat^®^ version 4 (Xunta de Galicia, Santiago de Compostela, Spain, and Pan American Health Organization) statistical software. To establish a qPCR cut-off, the prospectively recruited patients from group 1 were used since they had provided the necessary information and their samples had been processed immediately for the microscopic visualization.

The cut-off was established using the receiver operating characteristic (ROC) curve to determine the highest sensitivity and specificity using microscopy results (TBO and/or DFA) as the reference test. The sensitivity, specificity, and positive and negative predictive values are reported for group 1.

We used Chi-squared and Mann–Whitney U-tests to explore differences in the frequency of PjP and colonization compared to non-infected by Pj. The prevalence ratio (PR) with a 95% confidence interval is reported.

## 3. Results

### 3.1. Diagnostic Validation of qPCR in BAL

We included 213 individuals: 158 immunocompromised patients, 35 with suspected lung cancer, and 20 healthy subjects without any indication for BAL, in which only OW was taken (Figure 1).

Among the immunocompromised patients, 79.7% were infected with the human immunodeficiency virus (HIV)/AIDS, 13.9% were transplant recipients, 3.2% had a connective tissue autoimmune disease and were taking immunosuppressants, and 3.3% had hematologic neoplasia. Majority of the HIV/AIDS patients with available CD4 counts were severely immunosuppressed (CD4 count: median, 53 cell/mm^3^, IQR 16–126; CD4 < 200 cell/mm^3^: 88.5%) (Table 2).

In total, 25 cases of PjP were diagnosed by TBO and/or DFA; of those, 24 cases (96%) were HIV-positive patients, and 1 (4%) had undergone a kidney transplantation. 

Among the immunocompromised patients, qPCR was positive in 72.2% of the BAL samples (114/158), and Ct values ranged from 13.99 and 36.99. Among patients with microscopically confirmed PjP, median Ct values were lower than the Pj negative cases by microscopy (18.16, IQR 16.47–21.55 vs. 33.80, IQR 30.90–35.42, *p* < 0.0001).

Based on the BAL sample ROC curves from immunocompromised patients (group 1), we determined that the Ct value of 24.53 was the best cut-off to discriminate between PjP disease and colonization (sensitivity 100% (CI 95% 98–100), specificity 97.7% (CI 95% 94.8–100), positive predictive value 89.3% (CI 95% 76.0–100), negative predictive value 100% (CI 95% 99.6–100), area under the curve (AUC) ROC 0.985, 95% CI 0.953 to 1.00) (Figure 2A). A Ct value lower than 24.53 was defined as PjP (confirmed or probable), while Ct values higher than the established threshold were considered to represent colonization.

This cut-off of 24.53 confirmed all of the microscopy-positive cases and identified three additional cases of PjP among the BAL of individuals from group 1. These three patients had a new HIV diagnosis, met the criteria of probable PjP, and had individual Ct values of 16.05, 16.66, and 18.36. None of them died during the follow-up.

Among the Pj negative microscopy samples (133 patients), three patients were considered as probable PjP as previously mentioned, and among the 130 remaining patients of group 1, 66.2% (86/130) were classified as colonized (Ct results between 25.03 and 36.99) (Figure 3A).

No PjP cases were detected by microscopy or qPCR in the BAL samples of patients with lung cancer, but 48% of individuals were found to be colonized (Ct results between 31.61 and 36.52) (Figure 3B).

### 3.2. Analysis of qPCR in OW Samples

qPCR was positive in 53 of the 158 OW samples from the immunocompromised patients, including 19 of the 25 microscopy-positive BAL samples (Figure 3B). The Ct values obtained from the 19 qPCR and microscopy-positive patients were lower than the negative sample values (Me 31.08; IQR 28.66–34.44 vs. Me 35.61, IQR 34.12–36.30, respectively, *p* ≤ 0.001), and as such, it was not possible to establish an acceptable cut-off for PjP diagnosis in the OW samples (AUC ROC: 0.815, 95% CI 0.704 to 0.926) (Figure 2B). Nine of fifty-three patients were positive by qPCR in the OW sample, although none were positive in the BAL. The correlation between the Ct values of the BAL samples and those of the OW samples was low (ρ = 0.43, *p* ≤ 0.001) (Figure 4).

In the cancer patient group, 6 of the 35 patients’ OW samples were qPCR-positive (17%, Ct values between 35.29 and 36.37), including those from three patients who also had a positive BAL (Ct range 35.29–35.76). qPCR was positive in 2 (Ct 31.5 and 35.32) out of 20 healthy individuals in their OW samples (Figure 3B).

### 3.3. Clinical Characteristics and Follow-Up of Immunosuppressed Patients, Colonized and Non-Infected Individuals

AIDS, CD4 count less than 200 cells/mm^3^, no prior history of prophylaxis with trimethoprim/sulfamethoxazole for >12 weeks, cough, dyspnea, pulsi oximetry < 90%, tachypnea, tachycardia, interstitial opacities, and ground-glass pattern were more prevalent in people with PjP compared to non-infected individuals. History of corticosteroid was more prevalent in people colonized by Pj compared to non-infected individuals (Table 3).

Regarding clinical outcomes, 32.1% of people with PjP required ICU admission, compared to 12.9% of people colonized with Pj and 9.1% with those who were negative for Pj. The in-hospital mortality was 14.3% in individuals with PjP, 11.6% in colonized individuals, and 9.1% in Pj negative individuals. People with Pj had higher mortality at 1 year of follow-up (39.3%), and there was no difference between people who were colonized by Pj (29.1%) and those who were Pj-negative (27.3%) (Long Rank test = 0.618) (Figure 5).

## 4. Discussion

Using our qPCR protocol and the threshold established herein, we were able to diagnose PjP and to discriminate between PjP and colonization in BAL samples of prospectively recruited patients with severe immunosuppression. In addition, we found that OW samples were not a useful sample type for discriminating between PjP and colonization, and we did not detect all PjP identified by TBO and DFA in BAL.

To date, we found 31 articles in PubMed that established a cut-off for molecular diagnosis of Pj using qPCR. Among them, 19 were retrospective [4,14,16,19,20,22,23,26,42,43,44,45,46,47,48,49,50,51], 9 were prospective [13,15,18,21,23,24,25,52,53], and 3 were mixed [12,54,55]. In 26 of them, authors accepted the diagnosis of PjP when one or more of the following criteria were present: clinical presentation suggestive of PjP, adequate response to PjP treatment, lung images suggestive of pneumonia caused by Pj, and/or a microscopy-positive Pj sample. Given that all patients were severely immunosuppressed and were at a higher risk for having one or more opportunistic infections, that there are no pathognomonic clinical or radiological presentations of PjP, and that an equivalent therapeutic response can be observed in pneumonia caused by other microorganisms, it is necessary to use microscopy to definitively diagnose PjP. However, despite these aforementioned points, only five studies used microscopy to reach their diagnoses [4,13,24,43,50].

Of the five studies, three were retrospective and two were prospective, and all of the populations were immunosuppressed (albeit due to different etiologies). Only two studies used ROC curves to establish a cut-off [13,24]. Fillaux et al. amplified the multicopy gene MSG from 400 fresh BAL samples, and established a Ct cut-off value of <22 as positive for Pj and >28 as negative, while those between this range were identified as indeterminant [24]. Conversely, Moodley et al., used the same mtLSU gene that was used in our study; however, they targeted a different region of the gene and established a different cut-off value (Ct 38.19). In their study, they evaluated 305 patients using Pj immunofluorescence and found 156 positives, 138 negatives, and 11 indeterminant cases. When they compared those results with qPCR, they found that qPCR detected 204 positive cases (41 false positives) and 101 negatives (4 false negatives), and had an AUC of ROC curve of 0.92 [13]. However, the results of Moodley et al. cannot be directly compared to the study herein since their patients had a different type of immunosuppression, the qPCR technique amplified a different region of mtLSU, and they used a probe-based detection protocol.

Overall, these studies highlight the heterogeneity of previously published results, and therefore, the difficulties associated with diagnosing Pj infections using molecular techniques. Consequently, our study, which identifies a new qPCR that allows differentiation between colonization and infection with high sensitivity and specificity (>98% for both) in prospectively recruited patients, will serve as a stepping stone for developing a Pj diagnostic procedure that reflects current clinical practice, and should be evaluated in other studies and populations.

As previously mentioned, our qPCR detected 28/158 PjP cases in group 1, which was 10.7% more than microscopy alone. Since PJP diagnosis traditionally relies on visualization of the organism in respiratory samples, the three additional cases detected using our qPCR method could have been misinterpreted as PJP colonization had other PCR assays been used. However, in our study, we considered these cases to be PjP due to their new HIV diagnosis, their high fungal load, and the clinical diagnosis of PjP, including that the patient improved with treatment dose of PjP, with no other microbiological diagnosis that explained the cause of pneumonia, and these three patients were alive during the 1-year follow-up. This divergence from the microscopy findings may be attributed to the presence of co-infections, as two of the samples were also infected with *Mycobacterium avium* complex, a factor that could affect Pj visualization [56].

Furthermore, qPCR identified 86 cases of Pj colonization (54%) among our patient groups, all of which were from immunosuppressed individuals who had had respiratory symptoms, and were negative by microscopy. These findings match those observed previously in Pereira RM et al., which found Pj colonization rates of 44% among HIV-positive Brazilian participants with no respiratory symptoms. Moreover, they found that the lower the CD4 count, the higher the percentage of colonization [57].

Our cut-off point could also be validated in group 2, where 48% of patients with lung cancer (n = 17) had Ct values within the range of colonization, a finding which is consistent with that currently reported in literature (i.e., between 20–100% according to the patient’s clinical condition) [8,28].

Currently, it is unknown whether Pj colonization contributes to pneumonia development by itself or if it does so with other respiratory pathogens. Additionally, our understanding of how Pj interacts with other microorganisms, or under which conditions a colonized subject can become infectious, is incomplete. Therefore, it is necessary to conduct cohort studies supported by molecular epidemiological tools to evaluate these aspects, which could help increase our understanding regarding the role of Pj as a colonizer [18,27,58].

Regarding the OW samples, we were unable to establish a threshold to differentiate between disease and colonization; however, we were able to detect Pj. Additionally, OW Ct values had a low correlation with the Pj Ct values seen in BAL samples. It has been reported that during respiratory infections, there are discrepancies between respiratory samples [59,60,61], not only because of the dynamic of the infection, but also because respiratory samples are not homogeneous matrices due to the presence of mucus. As Pj often only passes transiently through the upper respiratory tract and replicates within the alveolar space, this finding was to be expected. In the OW samples, we found that 10% of the healthy individuals were colonized with Pj, a rate which has been reported previously in studies using nested PCR amplifying for the *mtLSU rRNA* gene [31,62,63,64]. Pj could also be a transient colonizer in the upper airway, as we previously reported in newborns and their mothers [65].

The main limitation of this study was that we could not establish a cut-off for people with severe immunosuppression that are non-HIV-positive, as suggested by previous researchers, because most of our patients were people living with HIV.

## 5. Conclusions

The qPCR threshold established herein allowed for reliable diagnosis of PjP and differentiation between PjP disease and colonization in BAL of immunocompromised patients with pneumonia. In contrast, OW samples were not useful in distinguishing between disease and colonization. Our qPCR should be evaluated in other populations to validate our findings.

## Figures and Tables

**Figure 1 jof-07-01036-f001:**
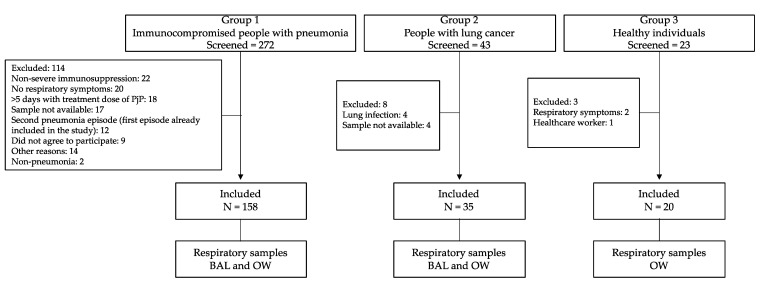
Flowchart of the study, including the type of respiratory samples analyzed in each group. BAL: Bronchoalveolar lavage. OW: Oral wash. PjP: *Pneumocystis jirovecii* pneumonia.

**Figure 2 jof-07-01036-f002:**
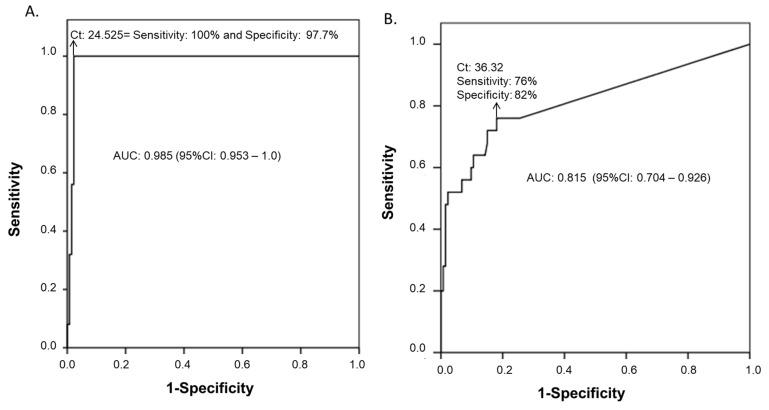
ROC curve analysis of the BAL and OW samples from immunosuppressed patients. With the arrow, the Ct value corresponding to the cut-off at which the best value of sensitivity and specificity for a diagnostic test is indicated when it is compared with BAL microscopy, according to sample type: (**A**) BAL; (**B**) OW.

**Figure 3 jof-07-01036-f003:**
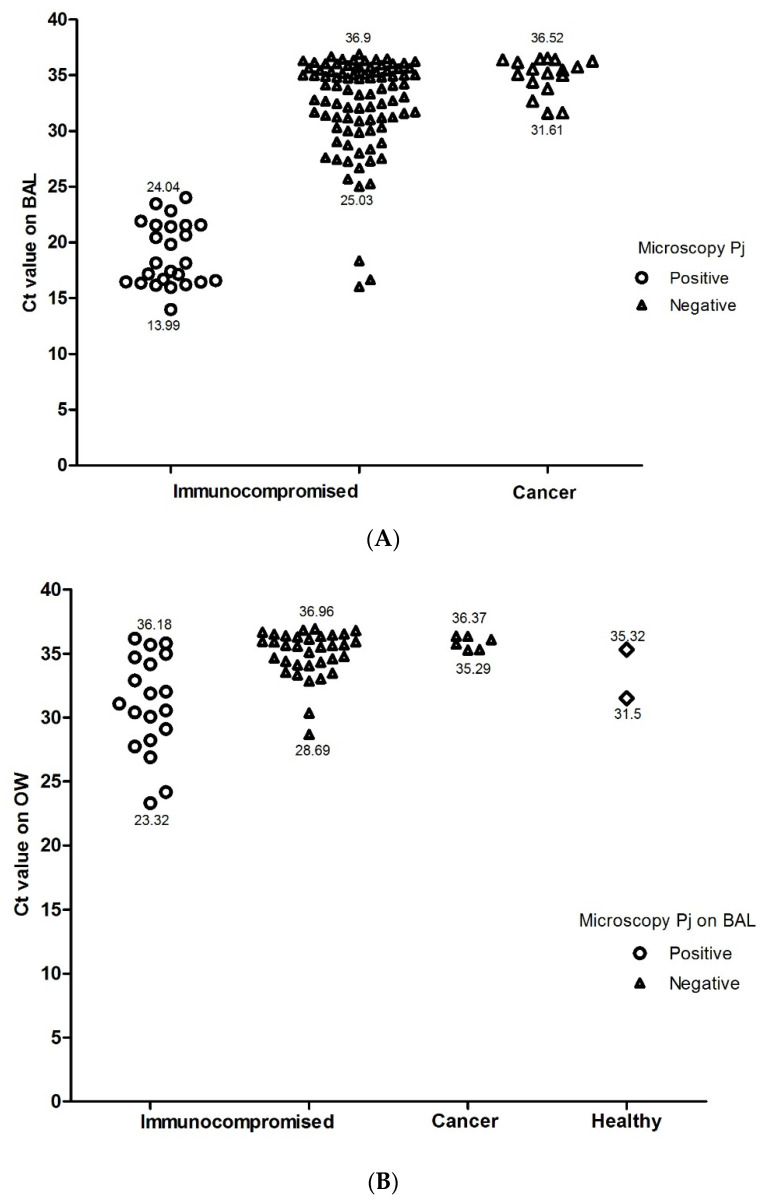
Ranges of the Ct values for the BAL (**A**) and OW (**B**) samples, according to microscopy results in the studied groups. The scatterplot shows Ct values obtained in each respiratory sample according to the group and microscopy Pj results on BAL. In the healthy group, BAL was not performed. The numbers indicate the lower and upper Ct values.

**Figure 4 jof-07-01036-f004:**
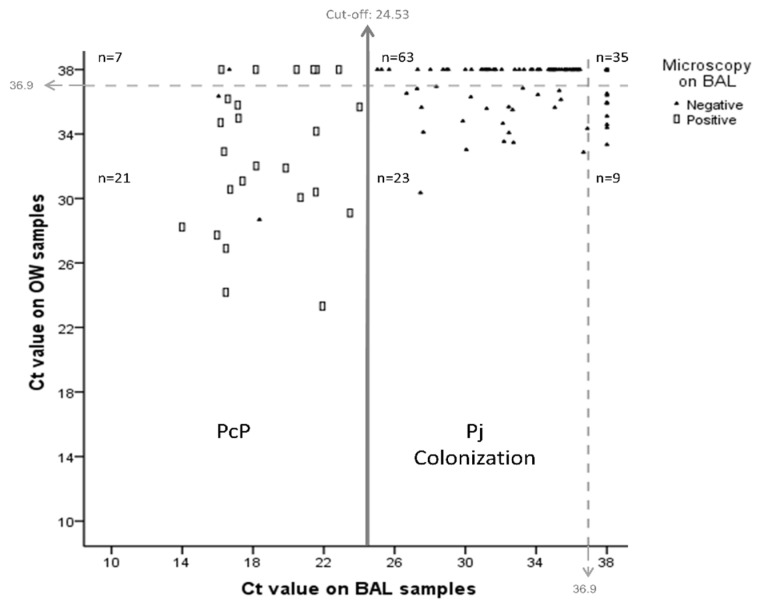
Scatterplot displaying the relationship between the Ct values for the BAL and OW samples in 158 patients. The continuous vertical line separates PjP disease from colonization, according to the cut-off established in BAL (ct = 24.53). Dashed lines indicate the maximum Ct value (36.9) above which qPCR was considered negative. In each case, the microscopy result for *P. jirovecii* is specified in the BAL samples (□positive, ▲negative).

**Figure 5 jof-07-01036-f005:**
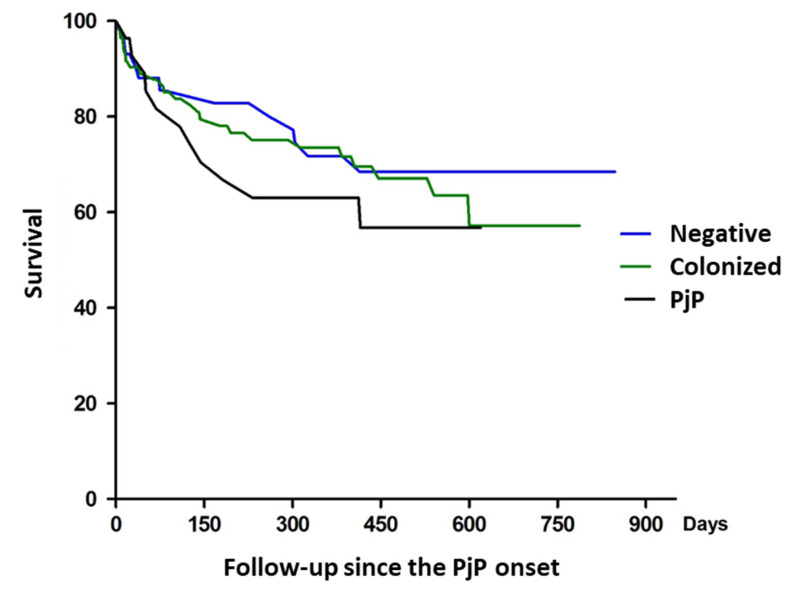
Survival of people with severe immunosuppression (28 with *Pneumocystis jirovecii* pneumonia (PjP), 86 colonized by Pj and 44 non-infected by Pj) at 1 year of follow-up.

**Table 1 jof-07-01036-t001:** Eligibility criteria of the 3 different groups included in the study.

	Group 1 Immunocompromised People with Pneumonia	Group 2 People with Suspicion of Lung Cancer without Pneumonia	Group 3 Healthy Blood Donors
Inclusion criteria	Individuals must have met all of the following criteria: ≥18 years Have at least one of the following severe immunosuppression conditions: primary immunodeficiencies, AIDS (with CD4+ T lymphocyte count ≤200/μL or category C, or CD4+ had decreased ≥50% in the past 6 months), solid organ or bone marrow transplant, hematologic neoplasia, CD4+ T lymphocyte count ≤200/μL from other causes distinct from AIDS, and other diseases that cause cellular immunity deterioration such as cancer and autoimmune diseases of the connective tissue, that required treatment with immunosuppressant drugs such as prednisone, 0.3 mg/k/d or its equivalent, for more than 2 weeks. People with pneumonia defined as the presence of respiratory symptoms (at least one of the following symptoms: cough, dyspnea, pleuritic pain, hemoptysis) or fever ≥37.8 °C, and with at least one of the following: abnormal breath sounds on lung auscultation or pulmonary opacities in the chest X-ray or high-resolution computed tomography. Underwent bronchoscopic bronchoalveolar lavage (BAL). Agreed to participate in the study.	Individuals must have met all of the following criteria: ≥18 years BAL was obtained as part of the study to confirm lung cancer diagnosis Agreed to participate in the study	Individuals must have met all of the following criteria: Immunocompetent and between 18 and 65 years old No symptoms of acute respiratory tract infection at the time of OW sampling or any other type of respiratory infection in the past month No heart disease or chronic pulmonary disease No pregnancy No immunosuppression: No corticosteroids or other immunosuppressing therapy for any reason in the past 3 months No hematological or invasive cancer in the past 5 years Diabetes, autoimmune disease, or granulocytopenia < 500 cell/mm^3^
Exclusion criteria	Those who received more than 5 days of treatment dose of PjP (not prophylactic treatment): trimethoprim/sulfamethoxazole (15/75 mg/kg/day) or clindamycin + primaquine. Oral wash not taken within 48 h of BAL.	Those with lung infection or antimicrobial treatment at the time of BAL Oral wash not taken within 48 h of BAL	Those who had been in a hospital in the past month for more than 4 h/day

**Table 2 jof-07-01036-t002:** Clinical and laboratory baseline features in all study groups.

Characteristics	Group 1: Immunocompromised Patients with Pneumonia n = 158	Group 2: Individuals with Lung Cancer n = 35	Group 3: Healthy Individuals n = 20
Male sex, n (%)	113 (71.5)	26/35 (74.3)	11 (55)
Age in years, median (IQR)	37 (29–45)	63 (58–70)	39 (21–50)
Respiratory symptoms, n (%)			0
Cough	118 (75.2)	24 (70.6)	
Dyspnea	63 (40.1)	19 (55.9)	
Pleuritic pain	27 (17.2)	6 (17.6)	
Hemoptysis	14 (8.9)	9 (26.5)	
Fever ≥ 38 °C	121 (77)	13 (38.2)	
Pleural effusion	24 (15.3)	14 (41.2)	
Past medical history (clinical condition at baseline), n (%)	158 (100)	0	0
AIDS	126 (79.7)		
CD4, cells/mm^3^, median (IQR)	53 (16–126)		
CD4 < 200, cells/mm^3^, n (%)	85 (88.5)		
Transplanted	22 (13.9)		
Connective tissue disease	5 (3.2)		
Hematologic malignancies	5 (3.2)		
Total leukocytes cell/mm^3^, median (IQR)	5760 (3700–7800)	11,860 (8300–14,800)	
PMN	4117 (2300–6175)	8449 (5218–10,962)	
Lymphocyte	835 (490–1460)	2075 (1540–2772)	
Pulmonary lung infection	**	0	N/A
*Pneumocystis jirovecii*	25 (16)		
*Mycobacterium tuberculosis*	30 (19)		
*Cryptococcus neoformans*	6 (4)		
*Histoplasma capsulatum*	4 (2.5)		
Bacteria	11 (7)		
Unknown	84 (53)		
ICU admission	24 (15)	0	N/A
In-hospital mortality	18 (11)	1 (2.9)	N/A

** In group 1, there were 6 patients with mixed infection: 1 with Pj and *M. tuberculosis* infection, 1 Pj + bacteria, 1 *M. tuberculosis* + bacteria, 1 *H. capsulatum* + *C. neoformans*, 1 Pj + bacteria + *M. tuberculosis*, 1 *M. tuberculosis* + nontuberculous mycobacteria. N/A: Not applicable. IQR: Inter-quartile range.

**Table 3 jof-07-01036-t003:** Factors associated with *Pneumocystis jirovecii* (Pj) in 158 patients with severe immunosuppression with pneumonia.

Characteristics	PjP Ct < 24.5 N= 28	Colonized Ct 24.5–37 N= 86	Non-Infected Ct > 37 N= 44	PjP vs. Colonized Individuals * PR; (CI 95%)	PjP vs. Non-Infected Individuals ** PR; (CI 95%)	Colonized vs. Non-Infected Individuals ** PR; (CI 95%)
Males, *n* (%)	20 (71.4)	62 (72.1)	31 (70.5)	0.976 (0.479–1.99)	1.01; (0.75–1.37)	1.02; (0.81–1.29)
Past medical history (clinical condition at baseline)						
AIDS, *n* (%)	27 (96.4)	64 (74.4)	35 (79.5)	6.82 (0.98–47.62)	1.212; (1.03–1.43)	0.94; (0.77–1.14)
Prior ART treatment, *n* (%)	2/13 (15.4)	19/45 (42.2)	17/30 (56.7)	2.92 (0.71–12.06)	0.27; (0.07–1.01)	0.74; (0.47–1.18)
CD4 < 200, cells/mm^3^, *n* (%)	20/20 (100)	41/46 (89.1)	24/29 (82.8)	NE	1.21; (1.02–1.43)	1.08; (0.89–1.31)
HIV viral load > 50.000, *n* (%)	9/13 (69.2)	21/32 (65.6)	18/22 (81.8)	1.13 (0.41–3.06)	0.85; (0.56–1.28)	0.80; (0.58–1.10)
Transplanted, *n* (%)	1 (3.6)	17 (19.8)	4 (9.1)	0.23 (0.03–1.54)	0.39; (0.05–3.34)	2.17; (0.78–6.07)
Hematologic malignancies, *n* (%)	0	3 (3.5)	2 (4.5)	NE	0	0.77; (0.13–4.43)
Systematic lupus erythematosus, *n* (%)	0	2 (2.3)	3 (6.8)	NE	0	0.34; (0.06–1.97)
Prior corticosteroid therapy *n*/N (%)	0/1	5/18 (27.8)	4/5 (80)	0.18 (0.02–1.27)	0	0.35; (0.15–0.82)
Neutrophils < 1500 cells/mm^3^	0/1	2/21 (9.5)	2/9 (22.2)	0.39 (0.06–2.54)	0	0.43; (0.07–2.59)
No history of prophylaxis with trimethoprim/sulfamethoxazole for >12 weeks	28/28 (100)	70/85 (82.4)	33/44 (75)	NE	1.33; (1.12–1.58)	1.09; (0.91–1.34)
**Respiratory symptoms, *n* (%)**						
Cough	27/28 (96.4)	59/85 (69.4)	32/44 (72.7)	8.47; (1.20–59.49)	1.33; (1.09–1.61)	0.95; (0.76–1.20)
Dyspnea	24/28 (85.7)	26/85 (30.6)	13/44 (29.5)	7.56; (2.80–20.37)	2.90; (1.79–4.69)	1.03; (0.59–1.81)
Pleuritic pain	3/28 (10.7)	5/85 (5.9)	7/44 (15.9)	1.58; (0.60–4.10)	0.67; (0.19–2.39)	0.37; (0.12–1.10)
Chest pain	7/28 (25)	12/85 (14.1)	8/44 (18.2)	1.65; (0.82–3.32)	1.38; (0.56–3.37)	0.78; (0.34–1.76)
Hemoptysis	1/28 (3.6)	7/85 (8.2)	6/44 (13.6)	0.49; (0.08–3.13)	0.26; (0.03–2.06)	0.60; (0.22–1.69)
Fever ≥ 38 °C	23/28 (82.1)	65/85 (76.5)	33/44 (75)	1.31; (0.55–3.09)	1.09; (0.86–1.39)	1.02; (0.83–1.26)
Pulse oximetry < 90	12/21 (57.1)	8/60 (13.3)	6/30 (20)	4.07; (2.02–8.2)	2.85; (1.28–6.39)	0.67; (0.25–1.75)
Respiratory frequency > 20 breaths/min	22/28 (78.6)	26/85 (30.6)	18/43 (41.8)	4.97; (2.18–11.3)	1.88; (1.26–2.81)	0.73; (0.45–1.18)
Tachycardia	17/28 (60.7)	31/85 (36.5)	13/44 (70.6)	2.09; (1.08–4.05)	2.05; (1.19–3.54)	1.23; (0.72–2.11)
Lymphocytes < 750 cells/mm^3^	15/28 (53.6)	39/84 (46.4)	15/44 (34.1)	1.239 (0.65–2.36)	1.57; (0.91–2.69)	1.36; (0.85–2.18)
LDH > 450 UI/L (n = 118)	16/24 (66.7)	19/62 (30.6)	10/32 (31.2)	2.91 (1.40–6.06)	2.13; (1.19–3.84)	0.98; (0.52–1.85)
**Radiographic features, *n* (%)**						
Normal radiographic	4/28 (14.3)	26/81 (32.1)	12/44 (27.3)	0.44; (0.166–1.16)	0.53; (0.188–1.46)	1.18; (0.66–2.10)
Interstitial opacities	23/28 (82.1)	39/81 (48.2)	21/44 (47.7)	3.49; (1.43–8.49)	1.72; (1.21–2.45)	1.01; (0.69–1.48)
Pleural effusion	2/28 (7.1)	15/85 (17.6)	7/44 (15.9)	0.43; (0.11–1.66)	0.45; (0.10–2.01)	1.11; (0.49–2.52)
Ground-glass pattern	13/16 (81.3)	11/53 (20.8)	5/31 (16.1)	8.13; (2.56–25.75)	5.04; (2.18–11.63)	1.29; (0.49–3.36)
**Complications and clinical outcomes**						
Acute respiratory distress syndrome	9/28 (32.1)	7/85 (8.2)	2/44 ((4.6)	2.87; (1.59–5.19)	7.07; (1.65–30.36)	1.81; (0.39–8.36)
Pneumothorax	0/28 (0)	2/85 (2.4)	1/44 (2.3)	NE	0	1.04; (0.09–11.11)
ICU admission	9/28 (32.1)	11/85 (12.9)	4/44 (9.1)	2.20; (1.17–1.13)	3.54; (1.20–10.39)	1.42; (0.48–4.213)
Mechanical ventilation required	6/28 (21.4)	10/85 (11.8)	4/44 (9.1)	2.20; (1.17–4.13)	2.36; (0.73–7.62)	1.29; (0.43–3.89)
In-hospital mortality	4/28 (14.3)	10/86 (11.6)	4/44 (9.1)	1.19 (0.49–2.92)	1.57; (0.43–5.78)	1.28; (0.43–3.85)

Ct: cycle threshold. PR: prevalence ratio. CI: confidence interval PjP: *Pneumocystis jirovecii* pneumonia. ART: HIV antiretroviral therapy. * The reference group is individuals colonized with *Pneumocystis jirovecii*. ** The reference group is the non-infected by Pj. NE: Not estimated when one of the group has either no cases or all cases had PjP.

## Data Availability

The database is available upon request.

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
