# Peer review of "Is It Possible to Differentiate Pneumocystis jirovecii Pneumonia and Colonization in the Immunocompromised Patients with Pneumonia?"

_jof, 2021, doi:10.3390/jof7121036_

Round 1
Reviewer 1 Report
I read with interest the manuscript by Aguilar and colleagues entitled “Is it possible to differentiate pneumocystis and colonization in immunocompromised patients with pneumonia? “The introduction is well focused, references are relevant and sample size is substantial. However, the article has significant weaknesses:
Cutoff values for distinguishing PCP and colonization were first published in 2002 by Larsen and colleagues (DOI: 10.1128/JCM.40.2.490–494.2002) and publications on the topic were thereafter numerous. The study of Aguilar and colleagues lacks originality and is of limited interest. There is no argument for using their method rather than those previously published.
Abstract
What do the authors mean when they state that sensitivity of staining methods is variable (L19)?
Materials and methods
This section is quite confusing. Table 1 is not well-structured; there are multiple inclusion criteria in group 1 but it is not clear whether all or only some of the criteria are required for patient inclusion. Why the authors chose a minimal 5-day treatment to exclude patients? Why did they not consider TMP/SMX at preventive dosage or other anti-Pneumocystis drugs as exclusion criteria?
The prospective design of the study is missing in this section.
The PCR assay used in this study has some technical weaknesses: It would be more accurate to use absolute quantitation using a standard and to use quantitative concentration data instead of Ct. Moreover, a probe would be more appropriate than Sybr green. This is not easily comprehensible since a number of qPCR assays amplifying the same target (mtLSU rRNA) and using standards and probe have been previously published.
Results
Typo L158 (158 instead of 58)
From which group was the cutoff value determined? Only group 1 or groups 1 and 2? It is not clear.
L181. A number to 2 decimal places (24.53) is not appropriate for routine diagnosis. Do the authors mean that a Ct value just above the cutoff value (24.54 for example) is sufficient to exclude PCP diagnosis?
L185. The authors state that “a Ct value lower than 24.53 was defined as PCP”; this is not correct since specificity was assessed at 97.7% with corresponding positive predictive value of 89.3%. So, colonization cannot be strictly excluded below the value of 24.53. It is needed to determine a second cutoff value reaching 100% specificity and then to set an indeterminate zone between the two cutoff values.
The authors should consider HIV status since it has been suggested that fungal load is higher in HIV-infected patients (doi: 10.1164/ajrccm/140.5.1204) and that cutoff values are different between HIV-positive and HIV-negative populations (doi:10.1128/JCM.02072-15).
The authors mentioned 3 additional cases of PCP (L191). They should indicate whether clinical data were consistent with PCP diagnosis (clinical signs? Anti-Pneumocystis treatment? Outcome?)
158 patients were included in Group 1. As mentioned in Table 2, 25 cases of PCP were diagnosed by microscopic detection of P. jirovecii in BALF samples. So, there should be 133 patients with negative microscopic detection of the fungus. Why do the authors mention 133 patients (L194)?
L222-225. Somewhat surprisingly, some OW samples were qPCR positive and corresponding BALF samples were negative. This should be clarified.
Typo L231 (corticosteroid)
Table 3: Why were PCP group and colonization group not compared to each other? It would be useful to perform first three-group comparisons and then pairwise group comparisons.
Discussion
L241. The authors state that OW samples can be used for P. jirovecii detection using a qPCR assay. This is not fully correct since qPCR sensitivity on this kind of specimen is not efficient enough.
Author Response
November 3, 2021
Editor
Journal of Fungi
Dear Editor:
Thank you very much for considering our manuscript. We really appreciate the reviewers’ comments as all of them made us realized that we missed very important information in the methods and results, as well as, there were some sentences that were not clear. Thanks to those suggestions the current version of our paper improved very much.
Following the journal instructions, the reviewers can identify the modifications in track changes.
Below we answer each comment of both reviewers.
Reviewer 1.
- I read with interest the manuscript by Aguilar and colleagues entitled “Is it possible to differentiate pneumocystis and colonization in immunocompromised patients with pneumonia? “The introduction is well focused, references are relevant and sample size is substantial. However, the article has significant weaknesses: Cutoff values for distinguishing PCP and colonization were first published in 2002 by Larsen and colleagues (DOI: 10.1128/JCM.40.2.490–494.2002) and publications on the topic were thereafter numerous. The study of Aguilar and colleagues lacks originality and is of limited interest. There is no argument for using their method rather than those previously published.
Answer: We appreciate your comments. We think there are several aspects that make our paper relevant, but we want to highlight two: 1) Low- and middle-income countries have limited resources, therefore we have to develop or adapt, and evaluate “cheaper” conditions that allow us to improve the diagnosis of P. jirovecii keeping high standards of research. 2) We agree that there are numerous publications that aimed to develop or reproduce the diagnosis of P. jirovecii by PCR, as we acknowledged and included them in the introduction and discussion of our paper, however, to our knowledge none of the qPCR developed until now have been endorsed as a gold standard for the diagnosis of Pneumocystis jirovecii.
- Abstract: What do the authors mean when they state that sensitivity of staining methods is variable (L19)
Answer: Thanks, we meant the sensitivity ranges between 20% and 99%. The differences in the sensitivity depend on the “gold standard” and definitions that researcher used to compare the stains. We removed this sentence.
- Materials and methods: This section is quite confusing. Table 1 is not well-structured; there are multiple inclusion criteria in group 1 but it is not clear whether all or only some of the criteria are required for patient inclusion. Why the authors chose a minimal 5-day treatment to exclude patients? Why did they not consider TMP/SMX at preventive dosage or other anti-Pneumocystis drugs as exclusion criteria
Answer: Thank you very much. The reviewer is right, it was not clear, and we missed to include some very important definitions.
In summary, for group 1 (immunocompromised people with pneumonia) participants must have met all five inclusion criteria: 1) age, 2) severe immunosuppression conditions, 3) have pneumonia and abnormal breath sounds on lung auscultation or pulmonary opacities in the chest x-ray or high-resolution computed, 4) underwent for bronchoalveolar lavage (BAL), 5) agreed to participate in the study. The specific immunosuppression conditions and the definition of pneumonia are defined in Table 1.
We also clarify that the exclusion criteria were those who received more than 5 days of treatment dose of P. jirovecii pneumonia (not prophylactic dose), and when the oral wash was not taken within the 48 hours of BAL.
We did not consider excluding those with history of prophylaxis with TMP/SMX because unfortunately in these hospitals, most patients were not taken or were not prescribed with Pneumocystis jirovecii prophylaxis due to late HIV diagnosis (in those with HIV) or late consultation.
We chose 5 days of treatment dose of PjP because when we were doing the pilot study, we found that the stains (our gold standard: TBO and DFA) became negative after 5 days of treatment. We agree that this variable can affect the results, and for that reason, we considered it in the analysis.
Finally, in Colombia we only have TMS/SMX and clindamycin/primaquine in case of any adverse event or allergy to TMS/SMX, for P. jirovecii treatment and prophylaxis. We clarified this in the paper.
- The prospective design of the study is missing in this section.
Answer: Our apologies. We included the description of the follow-up and definitions in the methods section and included the clinical outcomes during follow-up in the results section.
- The PCR assay used in this study has some technical weaknesses: It would be more accurate to use absolute quantitation using a standard and to use quantitative concentration data instead of Ct. Moreover, a probe would be more appropriate than Sybr green. This is not easily comprehensible since a number of qPCR assays amplifying the same target (mtLSU rRNA) and using standards and probe have been previously published.
Answer: We used the Ct values because it is unclear the meaning of the number of copies of the mtLSU rRNA gene in P. jirovecii. Sesterhenn T, et al. [Mol Genet Genomics MGG. 2010;283: 63–72. doi:10.1007/s00438-009-0498-7] published that the mitochondrial genome of Pneumocystis carinii has one copy of the mitochondrial large subunit ribosomal RNA (mt LSU rRNA; rnl). However, in fungi, one mitochondrion contains multiple copies of its mitochondrial genome. “This mitochondrial genome is packaged into protein-DNA complexes. These structures are called nucleoids. For example, S. cerevisiae has 10 to 40 nucleoids per-cell” (Biochim Biophys Acta. 2014 Jul;1837(7):1039-46. doi: 10.1016/j.bbabio.2013.10.005). Each nucleoid contains several mtDNA (mitochondrial DNA) copies (50-200) (Trends Microbiol. 2010;18: 521–530).
Regarding SYBR Green, this method is cheaper and easy to use, and our main interest was to have a cost-effective method, with high sensitivity and specificity due to the limited resources that we have in low- and middle-income countries. There are research that show that SYBR Green real-time PCR delivers highly comparable results with precise data compared to TaqMan PCR and other high-density microarrays (Tajadini M, et al. Adv Biomed Res. 2014 Feb 28;3:85. doi: 10.4103/2277-9175.127998 ; BMC Genomics 2008, 9:328 doi:10.1186/1471-2164-9-328).
It is important to note that the use of probe does not guarantee a complete specificity and can also produce false negative results. Le Gal S, et al, described that there is a mutation, C210T of the mtLSUrRNA gene. “This newly described mutation at position 210 is located at the hybridization region of the probe, that is, between positions 202 and 221 (Figure 1). This single mutation observed in this P. jirovecii specimen probably prevents the binding of the TaqMan® MGB probe due to the high stringency conditions of the amplification (temperature of hybridization, 60â—¦C), the small size of the probe and its high specificity due to MGB labeling. All PCR assays requiring MGB probes can be confronted with a false negative result due to a previously undescribed punctual mutation on the probe hybridization region. Thus, the occurrence of a false-negative PCR result provides arguments for maintaining microscopic techniques combined to PCR assays to achieve PCP diagnosis, which is the usual practice in France.” (Med Mycol. 2017 Feb 1;55(2):180-184. doi: 10.1093/mmy/myw051).
We submitted all technical details of our qPCR protocol to the Journal Biology Methods and Protocols. Below we paste some technical details of that protocol that it might help to clarify the question of the reviewer:
“Primer design
A real-time PCR assay amplified a fragment of the mitochondrial large subunit ribosomal RNA (mtLSU rRNA) gene, that has shown to be the most sensitive in the molecular diagnosis of this fungus [23]. Due the failure to quantify properly with the primers used in the nested PCR owing to the formation of dimers, a new set of primers were designed. For this proposal, a consensus sequence was obtained from those primers that were reported in GenBank for the mtLSU rRNA gene of Pj or P. carinii sp. hominis, and with Primer 3 software [24], the primers PJ mtLSU Forw (5´-AAGGGAAACAGCCCAGAACA-3´) and PJ mtLSU Rev (5´- CTGTTTCCAAGCCCACTTCTT-3´) were selected, and purified by HPLC (Integrated DNA Technologies ®, Iowa, USA). In addition, a locked nucleic acid was included in reverse the primer (C). The SNPs reported previously in mtLSU rRNA gene, does not affect the binding of our primers. The target sequences of mtLSU rRNA gene used of qPCR published is compared on graph supplementary (Supp. 1).
qPCR assay
After evaluating different conditions, the amplifications were performed in a final reaction volume of 25 µL: 0.3 µM of each primer, 1X SYBR-green QuantiFast (QIAGEN®, Hilden, Germany) and 5 µL of DNA from the samples or plasmid. The amplification and qPCR measurements were made using the SmartCycler thermocycler (Cepheid®, California, USA). The optimal cycling conditions were as follows: 95°C for 10 minutes, followed by 37 cycles of 95°C for 30 seconds, 65°C for 30 seconds and 72°C for 20 seconds. All runs were completed with a melt curve analysis to confirm the specificity of the amplification and lack of primer dimers. A positive reaction was deemed when a threshold cycle (Ct) between 10 and 36.99 accompanied by a Tm value of 77°C ± 1°C were obtained.
Construction of the p346 recombinant plasmid.
The amplicon of 346 bp was obtained from the first round of nested PCR that were amplified in the mtLSU rRNA region [5] from a PcP patient with microscopy positive. The fragment was ligated to the pGEM®-Teasy plasmid (Promega®, Southampton, USA), according to the manufacturer’s instructions. The recombinant plasmid (p346) was purified with the Wizard® plus SV Miniprep DNA Purification System (Promega ®, Southampton, USA), and was linearized with the RsaI restriction enzyme (Fermentas Life Sciences®, York, UK). After, the concentration and purity of plasmid DNA was measured using NanoDrop 2000 (Thermo Fisher Scientific Inc, USA), and was serially diluted 1:10 to construct the standard curve from 109 to 1 copy/reaction. The linearized plasmids were stored at -80°C for later use.
qPCR efficiency and analytical sensibility
The efficiency, linearity, and limit of quantification (LoQ) were assessed using a standard curve constructed with plasmid DNA p346. Additionally, the efficiency for the positive reaction from each patient sample or plasmid, was determined with LinReg software [25]. This approach identifies the exponential phase of the reaction by plotting the fluorescence on a log scale. A linear regression was then performed, leading to the estimation of the efficiency of each PCR reaction.
Determination of analytical specificity.
The novel qPCR with SYBRgreen detection was checked to assure that this method only amplifies the 105bp fragment of mtLSU rRNA gene in silico and experimentally, according to the guidelines (CLSI, CAP and FDA). The specificity of the primers was determined in silico using the nucleotide Basic Local Alignment Search Tool (BLASTN analysis), and the quality of the design was evaluated by determining the probability of the formation of hairpin, homo- and heterodimers with the Oligoanalyzer IDTsoftware (http://www.idtdna.com/analyzer/Applications/OligoAnalyzer/).
Experimentally, the qPCR product was verified with electrophoresis in a 2% agarose gel, Tm in melting curve and with the sequence reaction analysis with Blastn. In addition, cross-reaction was evaluated with DNA from human blood and other pulmonary pathogens obtained from culture of reference and clinical strains: Nocardia asteroides, Enterobacter cloacae, Klebsiella pneumoniae, Pseudomonas aeruginosa, Staphylococcus aureus, Haemophilus influenzae, Escherichia coli, Legionella pneumophila, Mycoplasma pneumoniae, Mycobacterium tuberculosis, Candida albicans, Candida guilliermondii, Candida glabrata, Candida tropicalis, Cryptococcus neoformans, Aspergillus terreus, Paracoccidioides brasiliensis, and Histoplasma capsulatum.
Determination of precision (Accuracy)
For the intra-assay reproducibility, five replicates of one BAL sample and one OW sample from a patient positive for Pj staining were made. For the inter-assay evaluation, three independent assays were conducted with the standard curve, and the findings were verified with all curves made during all time samples processing. For inter-operator reproducibility, four different investigators processed in independent assays with three different concentrations of p346 [108, 106 and 104(2x103, 2x 105 and 2x107 copies/µL)], one BAL and one OW from the same patient and one negative control, (in independent assays). It is expressed as standard deviation (SD) and coefficient of variation (CV) of Ct.”
- Results Typo L158 (158 instead of 58)
Answer: Thanks, the typo was corrected.
- From which group was the cutoff value determined? Only group 1 or groups 1 and 2? It is not clear. L181. A number to 2 decimal places (24.53) is not appropriate for routine diagnosis. Do the authors mean that a Ct value just above the cutoff value (24.54 for example) is sufficient to exclude PCP diagnosis.
Answer: Regarding the cutoff, it was determined using the patients of group 1 (immunosuppressed patients with pneumonia, n= 158). We included this clarification in the methods section.
We used two decimals to be more precise because we were using cycle thresholds (Ct) values, and the change between units is in a logarithm scale.
About the cutoff value of 24.54. Thanks to the reviewer’s comment, we realized that we did not include the definition of what we considered a positive case of Pneumocystis jirovecii pneumonia, and for this reason it was tough to understand. Now we included as a specific section entitled: “2.5 Definitions.
Pneumocystis jirovecii pneumonia was defined as:
Confirmed: The patient has pneumonia as defined in Table 1 and had a positive result by TBO and/or DFA in the BAL sample.
Probable: A positive qPCR below the cutoff established as a threshold for PjP diagnosis, and the patient had an interstitial pneumonia (pneumonia as defined in Table 1 and bilateral interstitial infiltrates in the chest x-ray or ground-glass pattern in the high-resolution computed tomography), with a negative result by TBO and DFA in the BAL sample, with no other microbiological diagnosis that explained the pneumonia, and had clinical improvement with treatment dose of PjP.
The research team did not participate and were not involved in any clinical decision from the infectious disease specialists or pulmonologists in charge of the patient. The research laboratory delivered the TBO and DFA results to the clinicians in charge of the patients immediately after the BAL samples were processed. As the qPCR was under development, we did not provide any qPCR result to the clinicians.”
The result of the qPCR was not used for any clinical decision, it means that all patients, independently of our study, that have suspicion of pneumonia underwent for BAL and were studied for fungal, mycobacterial and bacterial etiology using culture and direct stains in blood and in the BAL samples, and from any other sample that the clinician considered. The protocol of BAL that we used were published in 2007 (Respir Med. 2007 Oct;101(10):2160-7. doi: 10.1016/j.rmed.2007.05.017).
Clinicians decided the treatment of the patient based on the clinical symptoms and findings, and the severity of the pneumonia episode. They modified the treatment based on the evolution of the patient and the availability of the microbiological and laboratory results. The qPCR tests were processed blinded to any microbiological tests. Only when all samples were processed by qPCR, we compared qPCR results with TBO and DFA results.
Finally, we included in the table 1 and results of the paper the microbiological results (in addition to Pj), the clinical outcomes at discharge from hospitalization and at one year of follow-up.
Regarding the cutoff itself to establish a diagnosis of Pneumocystis jirovecii pneumonia, we calculated the ROC curve based on the patients from group 1. As the next comment is related, we expand the explanation below to avoid repetition.
- The authors state that “a Ct value lower than 24.53 was defined as PCP”; this is not correct since specificity was assessed at 97.7% with corresponding positive predictive value of 89.3%. So, colonization cannot be strictly excluded below the value of 24.53. It is needed to determine a second cutoff value reaching 100% specificity and then to set an indeterminate zone between the two cutoff values. The authors should consider HIV status since it has been suggested that fungal load is higher in HIV-infected patients (doi: 10.1164/ajrccm/140.5.1204) and that cutoff values are different between HIV-positive and HIV-negative populations (doi:10.1128/JCM.02072-15)
Answer: Thanks for your comment. The cutoff of 24.53 was established to determine diagnosis of Pneumocystis jirovecii pneumonia. Below we pasted the full table from the ROC curve. It is important to mention that to establish the cutoff we took into account not only the specificity, also the sensitivity because sometimes there are tests with high specificity but low sensitivity, which means that the disease of interest would be missed.
As it can be seen in the table, a cutoff with a specificity of 100% (the first three rows of the table), accounts for a sensitivity of 8% or less, which it would imply that our PCR will miss 92% of P. jirovecii cases, something unacceptable for any diagnostic test.
The cutoff of 24.53 gives us a sensitivity of 100% (detected all cases diagnosed by TBO and DFA) with a very high specificity of 97.7%. The negative predictive value of 100% means that a person with a negative result does not have P. jiroveciipneumonia. Based on this result, the first conclusion is that the qPCR is useful to diagnose patients with P. jirovecii pneumonia.
Coordinates of the Curve |
||
Test Result Variable(s): BAL Ct values |
||
Positive if Less Than or Equal To |
Sensitivity |
1 - Specificity |
12,9900 |
0.000 |
0.000 |
14,9800 |
0.040 |
0.000 |
16,0100 |
0.080 |
0.000 |
16,1100 |
0.080 |
0.008 |
16,1900 |
0.120 |
0.008 |
16,2850 |
0.160 |
0.008 |
16,4100 |
0.200 |
0.008 |
16,4650 |
0.240 |
0.008 |
16,5250 |
0.280 |
0.008 |
16,6200 |
0.320 |
0.008 |
16,6850 |
0.320 |
0.015 |
16,9200 |
0.360 |
0.015 |
17,1550 |
0.400 |
0.015 |
17,2950 |
0.440 |
0.015 |
17,7850 |
0.480 |
0.015 |
18,1700 |
0.520 |
0.015 |
18,2700 |
0.560 |
0.015 |
19,1050 |
0.560 |
0.023 |
20,1550 |
0.600 |
0.023 |
20,5700 |
0.640 |
0.023 |
21,0500 |
0.680 |
0.023 |
21,4850 |
0.720 |
0.023 |
21,5600 |
0.760 |
0.023 |
21,5800 |
0.800 |
0.023 |
21,7550 |
0.840 |
0.023 |
22,3850 |
0.880 |
0.023 |
23,1650 |
0.920 |
0.023 |
23,7500 |
0.960 |
0.023 |
24,5250 |
1.000 |
0.023 |
25,1450 |
1.000 |
0.030 |
25,4750 |
1.000 |
0.038 |
26,1850 |
1.000 |
0.045 |
26,9750 |
1.000 |
0.053 |
27,2850 |
1.000 |
0.060 |
27,3800 |
1.000 |
0.068 |
27,4900 |
1.000 |
0.075 |
27,5700 |
1.000 |
0.083 |
27,8200 |
1.000 |
0.090 |
28,1900 |
1.000 |
0.098 |
28,5500 |
1.000 |
0.105 |
28,8350 |
1.000 |
0.113 |
28,9800 |
1.000 |
0.120 |
29,4450 |
1.000 |
0.128 |
29,9300 |
1.000 |
0.135 |
30,0250 |
1.000 |
0.143 |
30,1800 |
1.000 |
0.150 |
30,3250 |
1.000 |
0.158 |
30,6200 |
1.000 |
0.165 |
30,9550 |
1.000 |
0.173 |
31,0750 |
1.000 |
0.180 |
31,1700 |
1.000 |
0.188 |
31,2150 |
1.000 |
0.195 |
31,2350 |
1.000 |
0.203 |
31,3100 |
1.000 |
0.211 |
31,4800 |
1.000 |
0.218 |
31,6300 |
1.000 |
0.226 |
31,6950 |
1.000 |
0.233 |
31,8800 |
1.000 |
0.241 |
32,0850 |
1.000 |
0.248 |
32,1500 |
1.000 |
0.256 |
32,3150 |
1.000 |
0.263 |
32,4550 |
1.000 |
0.271 |
32,5650 |
1.000 |
0.278 |
32,6950 |
1.000 |
0.286 |
32,7550 |
1.000 |
0.293 |
32,9200 |
1.000 |
0.301 |
33,1500 |
1.000 |
0.308 |
33,2800 |
1.000 |
0.316 |
33,5050 |
1.000 |
0.323 |
33,7500 |
1.000 |
0.331 |
33,9350 |
1.000 |
0.338 |
34,0850 |
1.000 |
0.346 |
34,1100 |
1.000 |
0.353 |
34,1700 |
1.000 |
0.361 |
34,4600 |
1.000 |
0.368 |
34,7150 |
1.000 |
0.376 |
34,7400 |
1.000 |
0.383 |
34,7750 |
1.000 |
0.391 |
34,8150 |
1.000 |
0.398 |
34,8600 |
1.000 |
0.406 |
34,9150 |
1.000 |
0.414 |
34,9550 |
1.000 |
0.421 |
34,9850 |
1.000 |
0.429 |
35,0050 |
1.000 |
0.436 |
35,0200 |
1.000 |
0.444 |
35,0400 |
1.000 |
0.451 |
35,0600 |
1.000 |
0.459 |
35,1150 |
1.000 |
0.466 |
35,2200 |
1.000 |
0.474 |
35,2950 |
1.000 |
0.481 |
35,3550 |
1.000 |
0.489 |
35,4100 |
1.000 |
0.496 |
35,5000 |
1.000 |
0.504 |
35,6000 |
1.000 |
0.511 |
35,6300 |
1.000 |
0.519 |
35,6600 |
1.000 |
0.526 |
35,6950 |
1.000 |
0.534 |
35,7200 |
1.000 |
0.541 |
35,8000 |
1.000 |
0.549 |
35,8900 |
1.000 |
0.556 |
35,9700 |
1.000 |
0.564 |
36,0400 |
1.000 |
0.571 |
36,0650 |
1.000 |
0.586 |
36,1100 |
1.000 |
0.594 |
36,1850 |
1.000 |
0.602 |
36,2550 |
1.000 |
0.609 |
36,2900 |
1.000 |
0.617 |
36,3350 |
1.000 |
0.624 |
36,3950 |
1.000 |
0.632 |
36,4250 |
1.000 |
0.639 |
36,4500 |
1.000 |
0.647 |
36,5750 |
1.000 |
0.654 |
36,7900 |
1.000 |
0.662 |
37,4500 |
1.000 |
0.669 |
39,0000 |
1.000 |
1.000 |
Once we have this cutoff, then, we compared the qPCR results of patients from group 1 against people that we thought would be ‘colonized’ by P. jirovecii based on previous literature and that they did not have pneumonia or any infection, group 2 and group 3. None of the persons from group 2 and 3 had values below 24.53 (the diagnostic threshold), but some of them had positive qPCR results between 24.54 to 36.99.
Based on this finding, the second conclusion is that there are people with lung cancer and healthy asymptomatic people that are colonized by P. jirovecii.
And the third conclusion is that there are immunosuppressed patients, mostly people living with HIV, are colonized by Pj.
Regarding the cutoff based on the underlying condition of the patients, we had 32 people severely immunosuppressed with pneumonia that did not have HIV, and only 1 of them had PjP. With only one positive case of P. jirovecii in an immunosuppressed person who was non-HIV, we cannot determine a cutoff in this particular group.
In Medellín, most of our patients with Pj are people living with HIV, as it has been reported in previous research (Infectio 2012;16(Suppl 3):23-30. Doi: 10.1016/S0123-9392(12)70023-1 ; Respir Med. 2007 Oct;101(10):2160-7. doi: 10.1016/j.rmed.2007.05.017).
- The authors mentioned 3 additional cases of PCP (L191). They should indicate whether clinical data were consistent with PCP diagnosis (clinical signs? Anti-Pneumocystis treatment? Outcome?)
Answer: Thank you. We included a detailed description of these three patients in the results of the paper. The three patients had a new HIV diagnosis, had a positive qPCR below the threshold (Ct values of 16.05, 16.66 and 18.36, respectively), interstitial pneumonia (pneumonia as defined in Table 1: had bilateral interstitial infiltrates in the chest x-ray, two of them had ground-glass pattern in the high-resolution computed tomography), with no other microbiological diagnosis that explained the pneumonia, had clinical improvement with treatment dose of PjP (not prophylactic treatment, and None of them died during the follow-up.
- 158 patients were included in Group 1. As mentioned in Table 2, 25 cases of PCP were diagnosed by microscopic detection of P. jirovecii in BALF samples. So, there should be 133 patients with negative microscopic detection of the fungus. Why do the authors mention 133 patients (L194)?
Answer: We did not understand this comment. We could not find the 133 patients in L194, nor in that paragraph in the version that was submitted to the journal. If the comment is why we called them “patients”, is because all people in group 1 has severe immunosuppression with pneumonia, all of them were hospitalized and were studied to identify the etiology of pneumonia, and many of them had other pathogens different to Pj.
We also clarified the following paragraph in case that was the confused paragraph: “Among the Pj negative microscopy samples (133 patients), three patients were considered as probable PjP as previously mentioned, and among the 130 remaining patients of group 1, 66.2% (86/130) were classified as colonized (Ct results between 25.03 and 36.99) (Figure 3A).”
- L222-225. Somewhat surprisingly, some OW samples were qPCR positive and corresponding BALF samples were negative. This should be clarified.
Answer: We included in the discussion (L313 to 321) a potential explanation. We think that it is due to a transitory passage of the fungus through the airway. It has been reported that during respiratory infections there are discrepancies between respiratory samples (BMC Infect Dis 11, 329 (2011). https://doi.org/10.1186/1471-2334-11-329 ; Can Respir J 2018 Apr 3;2018:6283935.
doi: 10.1155/2018/6283935 ; Clinical Infectious Diseases, 2021;9:e352–e356), not only because of the dynamic of the infection, also because respiratory samples are not homogeneous matrices due to the presence of mucus. In addition, it might suggest colonization, but this finding needs further studies. We included the following paragraph in the discussion:
“In regards to the OW samples, we were unable to establish a threshold to differentiate between disease and colonization, however we were able to detect Pj. Additionally, OW Ct values had a low correlation with the Pj Ct values seen in BAL samples. It has been reported that during respiratory infections there are discrepancies between respiratory samples[59–61], not only because of the dynamic of the infection, also because respiratory samples are not homogeneous matrices due to the presence of mucus). As Pj often only passes transiently through the upper respiratory tract and replicates within the alveolar space, this finding was to be expected. In the OW samples, we found that 10% of the healthy individuals were colonized with Pj, a rate which has been reported previously in studies using nested PCR amplifying for the mtLSU rRNA gene[31,62–64]. Pj could also be a transient colonizer in the upper airway, as we previously reported in newborns and their mothers[65].”
- Typo L231 (corticosteroid)
Answer: Thanks, we corrected the typo.
- Table 3: Why were PCP group and colonization group not compared to each other? It would be useful to perform first three-group comparisons and then pairwise group comparisons.
Answer: Table 3 only contains the information for group 1. We did the comparison and added to the table 3. The new table contains the prevalence ratio and 95% confidence intervals of the comparison between PjP group vs colonized group.
- Discussion L241. The authors state that OW samples can be used for P. jirovecii detection using a qPCR assay. This is not fully correct since qPCR sensitivity on this kind of specimen is not efficient enough
Answer: Thanks. We meant Pj detection, but its interpretation (Positive for PjP, or colonization) is not conclusive. For this reason, we modified the sentence as follow: “In addition, we found that OW samples were not a useful sample type for discriminating between PjP and colonization, as well as, did not detect all PjP identified by TBO and DFA in BAL”.
Reviewer 2 Report
I read with interest the article by Aguilar and colleagues entitled “Is it Possible to Differentiate Pneumocystosis and Colonization in Immunocompromised Patients with Pneumonia? submitted for publication in Journal of Fungi.
Using a mtLSU rRNA gene qPCR protocol and analyzing BAL and OW samples the authors were able to establish the threshold to differentiate between PCP disease and colonization of immunocompromised patients with pneumonia in BAL but not in OW samples.
The study is well designed and executed, it is reeds easily and provides novel information of interest. I have only few comments
Major comments
- Study design. Since there is no follow-up of the patients (or in any case these data are not included in the manuscript) the name of “cohort study” is not appropriate. When reading Table 3, I also have doubts that the follow-up is prospective, since the data for “Corticosteroid therapy n/N (%)” is available only in 1, 18 and 5 individuals of groups 1,2 and respectively. Similar for other variables of the table as “HAART treatment“ or “Neutrophils variable <1500 cells/mm3”
- Study population. In group 1 patients, it should be specified whether all immunocomprimised patients with pneumonia were included or only those with availability of BAL and OW samples. In the second situation, the total number of patients diagnosed in the participating centers during the study period should be specified. Regarding group 3, what was the criterion to select the 20 individuals of the sample from the blood donor population between November 2010 and February 2011?.
3 Eligibility criteria. In groups 1 and 2, the availability of BAL and OW samples should be included as inclusion criteria. The inclusion criteria for group 1 could be simplified (for example: criteria 3 could be omitted and 7, 8 and 9 included within criterion 6).
- Patients with PCP. After calculating the qPCR threshold in BAL for patients with a microbiologically confirmed diagnosis of PcP, the authors included three other microcopy-negative subjects. I have my doubts that this is methodologically correct. In any case, the authors should specify in their results the clinical-radiological characteristics, treatment with anti-PCP drugs and the clinical evolution of these three patients.
- Since in one of the two articles published on the relationship between lung cancer and colonization by Pneumocystis, the latter is specifically associated with the small-cell subtype, it would be interesting for the authors to specify the lung cancer subtypes of colonized subjects.
- Regarding the usefulness of OW to detect colonization or PCP, some citations from previous studies are missing and in my opinion should be added:
Respaldiza N, et al. J Eukaryot Microbiol. 2006; 53 Suppl 1: S100-1. doi: 10.1111 / j.1550-7408.2006.00188.x.
Matos O, et al. Eur J Clin Microbiol Infect Dis. 2001 Aug; 20 (8): 573-5. doi: 10.1007 / s100960100563
Minor comments.
There are several abbreviations that are not described in the text the first time they appear:
-ct in the abstract (line 27)
-IQR (line 127, also in the foot of table 2
A type in line 158: 58 instead of 158
Author Response
November 3, 2021
Editor
Journal of Fungi
Dear Editor:
Thank you very much for considering our manuscript. We really appreciate the reviewers’ comments as all of them made us realized that we missed very important information in the methods and results, as well as, there were some sentences that were not clear. Thanks to those suggestions the current version of our paper improved very much.
Following the journal instructions, the reviewers can identify the modifications in track changes.
Below we answer each comment of both reviewers.
- I read with interest the article by Aguilar and colleagues entitled “Is it Possible to Differentiate Pneumocystosis and Colonization in Immunocompromised Patients with Pneumonia? submitted for publication in Journal of Fungi. Using a mtLSU rRNA gene qPCR protocol and analyzing BAL and OW samples the authors were able to establish the threshold to differentiate between PCP disease and colonization of immunocompromised patients with pneumonia in BAL but not in OW samples. The study is well designed and executed, it is reeds easily and provides novel information of interest. I have only few comments. Major comments:
Study design. Since there is no follow-up of the patients (or in any case these data are not included in the manuscript) the name of “cohort study” is not appropriate. When reading Table 3, I also have doubts that the followup is prospective, since the data for “Corticosteroid therapy n/N (%)” is available only in 1, 18 and 5 individuals of groups 1,2 and respectively. Similar for other variables of the table as “HAART treatment“ or “Neutrophils variable <1500cells/mm3
Answer: Our apologies, as we mentioned in comment X of the reviewer 1, we missed this information. We included in methods and results the information regarding the cohort study: definitions, follow-up, clinical outcomes at discharge and at one-year follow-up.
We realized that the way was written was misleading. Corticosteroid therapy meant “prior” corticosteroid therapy. In table 1 and table 2 we reported the number of patients with a specific immunosuppression based on the inclusion criteria, most of them had AIDS, some were transplanted, other had rheumatological conditions, and few of them had prior corticosteroid therapy. To avoid confusion, we included “Past medical history” as a subtitle within those conditions. We also clarify that Table 3 contains the information about group 1 exclusively, divided among those with PjP disease, colonization, and Pj negative individuals.
- Study population. In group 1 patients, it should be specified whether all immunocomprimised patients with pneumonia were included or only those with availability BAL and OW samples. In the second situation, the total number of patients diagnosed in the participating centers during the study period should be specified. Regarding group 3, what was the criterion to select the 20 individuals of the sample from the blood donor population between November 2010 and February 2011?.
Answer: Thanks. We corrected Figure 1 and now it includes the number of people screened, the number of people excluded and the reasons, including the lack of samples available.
Regarding group 3, it was very tough because people must have met all inclusion criteria that we established:
- Immunocompetent and between 18 and 65 years old
- No symptoms of acute respiratory tract infection at the time of OW sampling or any other type of respiratory infection in the last month
- No heart disease or chronic pulmonary disease
- No immunosuppression:
- No corticosteroids or other immunosuppressing therapy for any reason in the past 3 months
- No hematological or invasive cancer in the last 5 years
- Diabetes, autoimmune disease or granulocytopenia <500cell/mm3
- No pregnancy
And the person could not have this exclusion criterion: Those who had been in a hospital in the last month for more than 4 hours/day.
- Eligibility criteria. In groups 1 and 2, the availability of BAL and OW samples should be included as inclusion criteria. The inclusion criteria for group 1 could be simplified (for example: criteria 3 could be omitted and 7, 8 and 9 included within criterion 6).
Answer: Thank you very much. We organized the Table 1 to make it clearer as you suggested, as well as, we included one clarification that reviewer 1 suggested.
- Patients with PCP. After calculating the qPCR threshold in BAL for patients with a microbiologically confirmed diagnosis of PcP, the authors included three other microcopy-negative subjects. I have my doubts that this is methodologically correct. In any case, the authors should specify in their results the clinical-radiological characteristics, treatment with anti-PCP drugs and the clinical evolution of these three patients.
Answer: Thank you. We included the definitions of PjP in the methods, and we also included why we kept those three patients as PjP. The three patients had a new HIV diagnosis, had a positive qPCR below the threshold (Ct values of 16.05, 16.66 and 18.36, respectively), interstitial pneumonia (pneumonia as defined in Table 1: had bilateral interstitial infiltrates in the chest x-ray, two of them had ground-glass pattern in the high-resolution computed tomography), with no other microbiological diagnosis that explained the pneumonia, had clinical improvement with treatment dose of PjP, and none of them died during the follow-up.
- Since in one of the two articles published on the relationship between lung cancer and colonization by Pneumocystis, the latter is specifically associated with the small-cell subtype, it would be interesting for the authors to specify the lung cancer subtypes of colonized subjects.
Answer: We agree with your comment, it would be interesting to have that information, unfortunately, at the time we collected the information, we did not realize to register the full pathological diagnosis of the lung biopsy. We reviewed it at that moment, but we only collected “bronchogenic carcinoma as yes or no.” but not the sub-types.
- Regarding the usefulness of OW to detect colonization or PCP, some citations from previous studies are missing and in my opinion should be added:
Answer: Thanks for your suggestion, we added the references that you recommend in the discussion.
- Minor comments. There are several abbreviations that are not described in the text the first time they appear
-ct in the abstract (line 27)
-IQR (line 127, also in the foot of table 2)
- Results Typo L158 (158 instead of 58)
Answer: Thank you. We defined all abbreviations the first time that we used, as well as we defined IQR. We corrected the typo.
Thank you again to all reviewers because thanks to your comments our paper improved dramatically.
Sincerely,
All authors
Reviewer 3 Report
The manuscript analyzed the value of a real-time PCR assay in BAL and oropharyngeal washes in 158 immunosupressed patients nearly 80% were HIV+), cancer patients and healthy individuals. A Ct value cut-off was proposed for the immunosuppressed group. OW samples were not useful in distinguishing between colonization and infection. The study is of interest. The patient cohort is very good characterized. The used methods are state-of-the-art. The tables and figures are appropriate. I have only some minor concerns.
- Patients with lung infection or antimicrobial treatment at the time of BAL were excluded. This seems to be not realistic for patients on the ICU. Most patients on ICU receive antibiotic treatment. Please comment how this fact can influence the results and interpretation.
- Nearly 80% of the immunosuppressed patients were HIV+ or had AIDS with 88.5% having CD4 <200 cell/mm3. This highly immunosuppression has influence on the fungal load and therefore on the Ct value/cut-off. Please discuss this issue.
- Page 1, Line 44: Please add the following paper as reference for diagnosis of PCP - Curr Fungal Infect Rep DOI 10.1007/s12281-014-0188-8.
- Please add a limitation part in the discussion.
Author Response
November 3, 2021
Editor
Journal of Fungi
Dear Editor:
Thank you very much for considering our manuscript. We really appreciate the reviewers’ comments as all of them made us realized that we missed very important information in the methods and results, as well as, there were some sentences that were not clear. Thanks to those suggestions the current version of our paper improved very much.
Following the journal instructions, the reviewers can identify the modifications in track changes.
Below we answer each comment of both reviewers.
- The manuscript analyzed the value of a real-time PCR assay in BAL and oropharyngeal washes in 158 immunosupressed patients nearly 80% were HIV+), cancer patients and healthy individuals. A Ct value cut-off was proposed for the immunosuppressed group. OW samples were not useful in distinguishing between colonization and infection. The study is of interest. The patient cohort is very good characterized. The used methods are state-of-the-art. The tables and figures are appropriate. I have only some minor concerns. Patients with lung infection or antimicrobial treatment at the time of BAL were excluded. This seems to be not realistic for patients on the ICU. Most patients on ICU receive antibiotic treatment. Please comment how this fact can influence the results and interpretation.
Answer: Thanks to your suggestion we improved the Table 1 to be more specific.
We excluded people with lung infection or antimicrobial treatment for group 2 (people with lung cancer) and group 3 (healthy individuals). The reason is because those were the groups that we expected that could be colonized.
About patients from Group 1, we identified the eligible patients in the emergency room of the hospitals (usually within the first 24 hours of admission), for this reason we were able to trace the number of days of TMS/SMX. Because of the protocol that was published in 2007, most of all immunosuppressed patients with suspicion of pneumonia undergo for BAL as soon as possible. We also hired a full-time clinician that went every day to the emergency room to identify all people that was admitted to the hospital with suspicion of pneumonia, then she reviewed the clinical chart and if the patient met the inclusion criteria and any exclusion criteria, she enrolled the patient in the study. We included this sentence in the paper.
We agree that TMS/SMX prescribed at treatment dose might cause false negative results of the stains, for this reason, in our pilot study we identified that stains become negative after 5 days of TMS/SMX and therefore we used that criteria.
- Nearly 80% of the immunosuppressed patients were HIV+ or had AIDS with 88.5% having CD4. This highly immunosuppression has influence on the fungal load and therefore on the Ct value/cut-off. Please discuss this issue.
Answer: We agree with the reviewer. However, this is our epidemiological reality, most people with severe immunosuppression and pneumonia are HIV-infected individuals, sadly many of them have late HIV diagnosis, and many of those who knew their HIV status were not taken the antiretroviral medications. We included a paragraph that mention that 32 patients from group 1 had other type of immunosuppression but only one had PjP. With only one patient with PjP without HIV (renal transplantation), we cannot analyze separately this group.
- Page 1, Line 44: Please add the following paper as reference for diagnosis of PCP - Curr Fungal Infect Rep DOI 10.1007/s12281-014-0188-8.
Answer: Thank you, the reference was included.
- Please add a limitation part in the discussion.
Answer: We included that the main limitation in our study was: “The main limitation of this study was that we could not establish a cut-off for people with severe immunosuppression that are non- HIV, as suggested previous researchers, because most of our patients were people living with HIV”
Thank you again to all reviewers because thanks to your comments our paper improved dramatically.
Sincerely,
All authors
Round 2
Reviewer 1 Report
Although the manuscript has been improved, originality and interest are still really low.
Author Response
November 17, 2021
Editor
Journal of Fungi
Dear Editor:
Thank you very much for the opportunity to correct the paper. Below we answer the suggestions from the reviewer.
Reviewer 1.
- English language and style are fine/minor spell check required
Answer: We paid a professional English native speaker and professor that reviewed and corrected the English of our paper.
- Are the conclusions supported by the results? Must be improved
Answer: The conclusion of the abstract was modified as follow: “This new qPCR allowed for reliable diagnosis of PjP, and differentiation between PjP disease and colonization in BAL of immunocompromised patients with pneumonia.”
The conclusion of the paper was modified as follow: “The qPCR threshold established herein allowed for reliable diagnosis of PjP, and differentiation between PjP disease and colonization in BAL of immunocompromised patients with pneumonia. In contrast, OW samples were not useful in distinguishing between disease and colonized. Our qPCR should be evaluated in other populations to validate our findings.”
- Although the manuscript has been improved, originality and interest are still really low.
Answer: We appreciate your comment. The authors believe that this manuscript contribute to the body of evidence supporting the use of quantitative PCR for diagnosis of PjP. As we mentioned in our previous letter, we think there are several aspects that make our paper relevant, but we want to highlight two: 1) Low- and middle-income countries have limited resources, therefore we have to develop or adapt, and evaluate “cheaper” conditions that allow us to improve the diagnosis of P. jirovecii keeping high standards of research. 2) We agree that there are numerous publications that aimed to develop or reproduce the diagnosis of P. jirovecii by PCR, as we acknowledged and included them in the introduction and discussion of our paper, however, to our knowledge none of the qPCR developed until now have been endorsed as a gold standard for the diagnosis of Pneumocystis jirovecii.
Thank you again to all reviewers because their comments since the first round of review until now, helped us to improve our paper dramatically.
Sincerely,
All authors